# Diagnostic Value of the Peptest^TM^ in Detecting Laryngopharyngeal Reflux

**DOI:** 10.3390/jcm10132996

**Published:** 2021-07-05

**Authors:** Karol Zeleník, Viktória Hránková, Adéla Vrtková, Lucia Staníková, Pavel Komínek, Martin Formánek

**Affiliations:** 1Department of Otorhinolaryngology and Head and Neck Surgery, Faculty of Medicine, University Hospital of Ostrava, 708 52 Ostrava, Czech Republic; karol.zelenik@fno.cz (K.Z.); viktoria.hrankova@fno.cz (V.H.); lucia.stanikova@fno.cz (L.S.); pavel.kominek@fno.cz (P.K.); 2Department of Applied Mathematics, Faculty of Electrical Engineering and Computer Science, VSB—Technical University of Ostrava, 708 00 Ostrava, Czech Republic; adela.vrtkova@vsb.cz

**Keywords:** laryngopharyngeal reflux, gastroesophageal reflux disease, Peptest^TM^, 24-h multichannel intraluminal impedance-pH, Reflux Finding Score

## Abstract

Background: The Peptest^TM^ is a non-invasive diagnostic test for measuring the pepsin concentration in saliva, which is thought to correlate with laryngopharyngeal reflux (LPR). The aim of this study was to investigate the diagnostic value of the Peptest in detecting LPR based on 24-h multichannel intraluminal impedance-pH (MII-pH) monitoring using several hypopharyngeal reflux episodes as criterion for LPR. Methods: Patients with suspected LPR were examined with the Reflux Symptom Index (RSI), Reflux Finding Score (RFS), fasting Peptest, and MII-pH monitoring. We calculated the accuracy, sensitivity, specificity, positive predictive value (PPV), and negative predictive value (NPV) of the Peptest, RSI, and RFS based on the threshold of one and six hypopharyngeal reflux episodes. Results: Altogether, the data from 46 patients were analyzed. When one hypopharyngeal reflux episode was used as a diagnostic threshold for LPR, the accuracy, sensitivity, specificity, PPV, and NPV were, respectively, as follows: 35%, 33%, 100%, 100%, and 3%, for the Peptest; 39%, 40%, 0%, 95%, and 0%, for the RSI; and 57%, 58%, 0%, 96%, and 0%, for the RFS. The accuracy, sensitivity, specificity, PPV, and NPV of the Peptest for diagnosing gastroesophageal reflux disease (GERD) were 46%, 27%, 63%, 40.0%, and 48%, respectively. Conclusions: A positive Peptest is highly supportive of a pathological LPR diagnosis. However, a negative test could not exclude LPR.

## 1. Introduction

Laryngopharyngeal reflux (LPR) is often involved in the development of laryngeal, pharyngeal, rhinological, and otological conditions [1,2,3,4]. The precise diagnosis of LPR remains challenging, particularly in patients with mild to moderate symptoms, because the symptoms and findings are nonspecific, and an accepted gold standard does not exist, as every diagnostic method shows false positive and false negative results. Therefore, multiple methods are used for most patients. The most validated diagnostic tool, which is deemed to be the “gold standard”, is the 24-h multichannel intraluminal impedance-pH (MII-pH) monitoring tool, which provides useful information about the severity, number, and type (acid, nonacid, mixed, upright/recumbent) of hypopharyngeal reflux episodes. However, MII-pH monitoring might be inconvenient for some patients, is costly, and is not available at all institutions [1]. Moreover, the possibility that probe insertion might influence the LRP could not be completely excluded. Therefore, new non-invasive diagnostic approaches have been developed, including the detection of pepsin in saliva samples with the Peptest^TM^ (RD Biomed, Hull, UK) [5]. The Peptest measures the pepsin concentration in saliva, which is thought to correlate with recent (i.e., within hours) hypopharyngeal reflux episodes. The Peptest has been widely used around the world as a non-invasive, economic, and easy method for diagnosing LPR. However, its diagnostic value has been questioned, and currently, it is an issue of ongoing debate. In fact, no single study has evaluated the diagnostic value of the Peptest compared to that of MII-pH monitoring based on the LPR diagnostic criterion, which is the number of hypopharyngeal reflux episodes detected.

The present study aimed to investigate the diagnostic value of the Peptest compared to that of the gold standard LPR diagnosis based on MII-pH monitoring. To provide a more in-depth analysis, we used two different criteria for diagnosing LPR: the detection of one or six hypopharyngeal reflux episodes. Moreover, the diagnostic value of the Peptest was compared to those of the Reflux Symptom Index (RSI) and the Reflux Finding Score (RFS). Furthermore, we investigated the diagnostic value of the Peptest for diagnosing gastroesophageal reflux (GERD) based on a positive DeMeester score. These results were expected to provide a basis for using the Peptest for LPR screening in situations where performing the MII-pH is not possible.

## 2. Materials and Methods

This prospective study was performed in accordance with the Declaration of Helsinki, the requirements of good clinical practice, and all applicable regulatory requirements. It was approved by the Institutional Review Board and registered at Clinicaltrials.com, identifier: NCT03904758. Written informed consent was obtained from all participants before any procedure was initiated.

### 2.1. Patient Selection, Inclusion/Exclusion Criteria, and Basic Examination

Consecutive adult patients with symptoms and signs of LPR were prospectively recruited from April 2019 to December 2020. Exclusion criteria were as follows: history of head and neck cancer, patients with diagnosed esophageal motility disorder, patients with an upper respiratory infection in the prior 4 weeks, smokers, patients with an alcohol dependency, and patients that refused to stop anti-reflux medication for the purpose of the study.

Enrolled patients were asked to complete a questionnaire for the RSI. Then, a flexible laryngoscopy was performed to determine the RFS. For performing the RFS, we randomly selected one of two physicians experienced at diagnosing LPR.

### 2.2. Pepsin Sample Collection

The pepsin concentration was measured in saliva samples with the Peptest. Patients were instructed to keep their standard food and drink regimen the day before examination and fast overnight before examination. Patients provided a saliva sample (1–5 mL; throat sputum) at the office, the morning after an overnight fast, and directly before introducing the MII-pH probe. The saliva sample was collected in a 30 mL universal sample collection tube and immediately analyzed with a standardized procedure described previously [6].

### 2.3. Multichannel Intraluminal Impedance-pH Monitoring

Patients that were taking anti-reflux medication chronically were asked to stop taking the medication before the study as follows: proton pump inhibitor therapy was stopped for 1 week; H_2_ blockers were stopped for 48 h; and drugs containing CaCO_3_ were stopped for 1 day before the study. Furthermore, all patients were asked to maintain normal daily activities. We performed MII-pH monitoring with the Digitrapper pH-Z Testing System (Medtronic, Minneapolis, MN, USA). A VersaFlex LPR ZNID22 + 8R impedance catheter (Medtronic, Minneapolis, MN, USA) was equipped with pH sensors located at 0 cm (proximal) and 22 cm (distal), and eight impedance rings located at −3, −1, 1, 13, 15, 17, 20, and 23 cm from the proximal pH sensor. Before recording, the catheter was calibrated with buffer solutions at pH values of 4.0 and 7.0. The proximal pH sensor was placed in the hypopharynx in the retrocricoid region, 2 cm above the upper esophageal sphincter, with flexible laryngoscopy guidance. A Digitrapper Recorder (Medtronic, Minneapolis, MN, USA) was used to record the data. Patients were instructed to record the time they spent eating, drinking, and in a horizontal position. Two physicians experienced in the technique manually analyzed the tracings with AccuView Reflux Software (Medtronic, Minneapolis, MN, USA) and a standardized method [7].

A hypopharyngeal reflux event was defined as an episode that was detected by two impedance sensors in the hypopharynx. An acidic event consisted of a gaseous or liquid reflux with a pH ≤4.0. A non-acid event was a gaseous or liquid reflux with a pH >4.0. Accordingly, a group of non-acid reflux covers weakly acidic, neutral, and alkaline reflux events. We defined acid, non-acid, and mixed LPR according to the work of Lechien et al. [8], based on the ratio of the number of acid reflux episodes to the number of non-acid reflux episodes. Acid LPR was defined as a ratio >2; non-acid LPR was defined as a ratio <0.5; and mixed reflux LPR was defined as a ratio of 0.51 to 2.0 [8]. For this study, one hypopharyngeal reflux event and then 6 hypopharyngeal reflux episodes were considered as the threshold for pathological LPR (for more details, see Section 2.4 below).

### 2.4. Statistical Analysis

Demographic parameters are expressed as the median and interquartile range (IQR), the absolute frequency, or the relative frequency (%). The significance of differences between groups was tested with the Mann–Whitney test or the chi-square test of independence for contingency tables. According to a study by Hoppo et al., the LPR diagnosis was first based on the occurrence of ≥1 hypopharyngeal reflux episode detected with MII-pH [7]. The diagnostic accuracies of the Peptest, RSI, and RFS were evaluated in terms of the accuracy, sensitivity, specificity, positive predictive value (PPV), and negative predictive value (NPV) with corresponding confidence intervals. Then, to evaluate the diagnostic value of the Peptest with more restrictive criteria, we performed the same calculations for the Peptest, RSI, and RFS, but the diagnosis was based on a threshold of six hypopharyngeal reflux episodes detected with MII-pH monitoring. This latter threshold was based on a previous study by Formánek et al. [9]. Moreover, we calculated the evolution of the accuracy, sensitivity, specificity, PPV, and NPV of the Peptest over a span of thresholds that ranged from 1 to 70 hypopharyngeal reflux episodes. Finally, the diagnostic value of the Peptest in diagnosing GERD was determined based on the threshold for the DeMeester score. The significance level was set to 0.05, and the statistical analysis was performed using R software, version 4.0.3 (R foundation, Vienna, Austria).

## 3. Results

We enrolled 52 consecutive patients with suspected LPR in the study. Six patients were excluded (four of them did not tolerate the MII-pH probe, and in two patients, the record of MII-pH was broken and could not be used), and 46 were analyzed (Table 1).

When one hypopharyngeal reflux episode was used as the diagnostic threshold for LPR, the accuracy, sensitivity, specificity, PPV, and NPV for the Peptest (Table 2) were as follows: 35%, 33%, 100%, 100%, and 3%, respectively. In comparison, the results for the RSI were as follows: 39%, 40%, 0%, 95%, and 0%, respectively; and the results for the RFS were as follows: 57%, 58%, 0%, 96%, and 0%, respectively (Table 2). When the diagnostic threshold for LPR was six hypopharyngeal reflux episodes, the Peptest accuracy, sensitivity, and NPV increased slightly, and its specificity and PPV decreased slightly (Table 3). The RFS had a higher sensitivity than that of the Peptest when either one or six hypopharyngeal reflux episodes were used as the diagnostic threshold for LPR. When the LPR threshold was changed from 1 to 70 episodes, the best accuracy, sensitivity, specificity, PPV, and NPV for the Peptest were observed with an LPR threshold of 16 episodes detected with MII-pH. However, despite this, the results were only moderate (Figure 1). The diagnostic accuracy, sensitivity, specificity, PPV, and NPV for the Peptest in diagnosing GERD were as follows: 48%, 27%, 63%, 40%, and 48%, respectively (Table 4).

## 4. Discussion

Many medical specialists (e.g., general practitioners, allergists, chest physicians, gastroenterologists, otolaryngologists, and others) widely use the Peptest around the world, which highlights the importance of this study. To date, this study was the first to compare the diagnostic value of the Peptest to that of MII-pH monitoring based on appropriate diagnostic criteria for LPR.

Previous studies compared the diagnostic value of the Peptest to those of the RSI and RFS [10,11,12]. Barona et al. found 98% specificity and 40% sensitivity for the Peptest when saliva samples were collected from fasting subjects. In studies that performed two Peptest examinations (one fasting and one an hour after the main meal), the Peptest showed 95% specificity and 48% sensitivity compared to the RSI. However, studies have shown that the RSI was not specific for LPR, and its diagnostic weaknesses have been discussed repeatedly [1,13]. Therefore, a comparison between the Peptest and the RSI cannot be taken as sufficient evidence of the diagnostic value of the Peptest. Similar results were found when the Peptest was compared to the RFS [12]. However, the RSF is also nonspecific, and laryngeal signs cannot be used to determine reflux changes in many situations (e.g., in smokers, after radiation, after an upper airway infection, etc.).

Other studies evaluated the diagnostic value of the Peptest compared to that of MII-pH or pH monitoring [6,14,15]. However, all those studies used the diagnostic criteria for GERD, not LPR. Dy et al. compared the MII-pH and the Peptest with criteria for GERD in a pediatric population. They stated that “pepsin lacks sensitivity as a diagnostic tool for evaluating extraesophageal reflux disease” [14]. Similarly, other authors employed the “gastroenterological” methodology when performing MII-pH monitoring. In that methodology, the proximal sensor was positioned 5 cm above the level of lower esophageal sphincter [6,15]. Nevertheless, it has been repeatedly shown that GERD and LPR are different diagnoses, and a GERD diagnosis does not prove or exclude LPR [1]. Therefore, it is not reasonable to investigate the diagnostic value of the Peptest in diagnosing LPR by comparing it to GERD MII-pH criteria.

Other studies that used MII-pH monitoring to evaluate hypopharyngeal reflux episodes (LPR criteria) employed a method other than the Peptest for detecting pepsin in saliva [16,17]. Na et al. performed ELISA to examine the total pepsin levels in saliva collected upon waking. They found that pepsin levels were significantly higher in patients with LPR symptoms and demonstrated at least one episode of proximal esophageal reflux during MII-pH monitoring. Furthermore, they found that the average pepsin level upon waking was higher than that measured at any other time. They concluded that measuring pepsin levels in the saliva upon waking may be a useful method for diagnosing LPR [16]. A similar study was conducted by Kimara et al., who performed MII-pH monitoring by placing the proximal sensor above the level of the upper esophageal sphincter. Those authors observed a significant correlation between salivary pepsin levels measured (with ELISA) in waking samples and MII-pH measurements [17]. Consistent with the study conducted by Na et al., they found that pepsin was most frequently detected in the specimen collected upon waking [17]. Although those results are interesting and promising, they did not predicate the diagnostic value of the Peptest.

A meta-analysis conducted by Wang et al. included some studies that compared signs (RSI) and symptoms (RFS), which were not specific for LPR, and other studies that used MII-pH or pH monitoring with diagnostic criteria for GERD, not LPR. Therefore, based on those pooled data, the sensitivity of 64% and specificity of 68% found for the Peptest in their meta-analysis were burdened with significant bias; consequently, the results of that meta-analysis must be interpretated with great caution [18].

The best-designed study on the subject of LPR was conducted by Bobin et al. Those authors concluded that they could not identify any significant associations between the RFS, the key symptoms observed during the test period, the MII-pH findings (with LPR methodology), the gastrointestinal endoscopy characteristics, or the pepsin concentrations in saliva samples determined with the Peptest [19]. Some “raw” data for testing the diagnostic value of the Peptest compared to that of MII-pH are missing. Consequently, we could not compare results between that study and the present study.

Another interesting study, conducted by Lechien et al., examined patients with the Peptest and with the MII-pH monitoring based on correct LPR methodology [20]. Those authors divided the patients into positive and negative Peptest groups. They concluded that the occurrence of pharyngeal reflux events during MII-pH monitoring was similar between groups. However, the diagnostic value of the Peptest was not stated [20].

To the best of our knowledge, no previous study has compared the Peptest to MII-pH monitoring based on diagnostic criteria for LPR (i.e., the number of hypopharyngeal reflux episodes) and provided data on diagnostic value of the Peptest. Many factors can influence the results of a comparison between the Peptest and MII-pH monitoring, but two factors are most important: first, the MII-pH protocol, probe placement, and criteria; second, the time of saliva collection for the pepsin analysis and diet within a 24 h period before sample collection [21,22].

To date, there is no consensus about the best time for collecting the saliva sample. In fact, there is significant heterogeneity among studies concerning the timing of saliva collection, and the results differ among the different studies [21,22]. However, well-designed studies from the last few years have supported the notion that saliva for pepsin examinations should be collected upon waking, and we used this methodology in the present study [16,17,20,22]. Regarding the diet within the 24 h period before saliva collection, it was confirmed by Lechien et al. that the saliva pepsin concentration was significantly associated with foods and beverages consumed during the evening dinner and during the testing period [22]. In our study, patients were instructed to have their standard food and drink regimen the day before examination and during MII-pH study to determine whether their standard diet and habits lead to LPR. We are aware that this approach can lead to some bias. On the other hand, information provided by tests reflect the “real situation” of patients.

There is a relatively strong consensus regarding the MII-pH protocol for diagnosing LPR. It has been shown that LPR can be diagnosed with the highest degree of exactness when the upper impedance and pH probes are positioned above the level of the upper esophageal sphincter and the number of hypopharyngeal LPR episodes is counted. However, a strong consensus has not been reached about how to interpret the hypopharyngeal data (extraesophageal reflux episodes) provided by MII-pH monitoring [1]. The widely accepted criterion for diagnosing pathological LPR is at least one hypopharyngeal reflux episode detected with MII-pH monitoring, as described by Hoppo et al. [7]. That threshold was established by examining 34 healthy individuals. On the other hand, it has been shown that some pharyngeal reflux episodes occur in healthy individuals. Oelschlager et al. reported a median of five pharyngeal reflux episodes in 10 asymptomatic controls during 24 h pH monitoring [23]. Additionally, Zerbib et al. detected 32 pharyngeal reflux events during 24 h pH monitoring in 12 healthy subjects, and 12 pharyngeal reflux events occurred in 1 subject. Nevertheless, the median number of pharyngeal reflux events was zero [24]. These discrepancies clearly demonstrated the challenges involved in clinical interpretations of MII-pH data in the diagnosis of LPR. Thus, we need accurate, reproducible diagnostic criteria and a broader consensus.

Accordingly, in the present study, we evaluated the diagnostic value of the Peptest with more than one criterion. We employed two different LPR thresholds: one and six hypopharyngeal reflux episodes, as described by Hoppo et al. [7] and Formánek et al. [9], respectively. Both analyses showed that the Peptest had high PPV but low sensitivity and NPV.

There are some unresolved issues concerning the role of pepsin examinations in saliva. First, it has been demonstrated that pepsin may enter epithelial cells and become reactivated in the Golgi system, where the pH is 5.0 [19,25]. The reactivation of pepsin may lead to mitochondrial and Golgi complex damage and, subsequently, cell destruction [19,25]. Therefore, the saliva pepsin concentration might not reflect the actual concentration of active pepsin in laryngopharyngeal tissues. In other words, only a limited proportion of harmful pepsin might be measurable with the Peptest. On the other hand, we hypothesize that the presence of pepsin in saliva might reflect relatively severe reflux, because it is unlikely that all extracellular pepsin is immediately internalized into mucosal cells. Second, pepsin is probably only one of many gastrointestinal enzymes (trypsin, lipase, and bile salts) that can cause LPR symptoms. Its role in the development of LPR must be studied more in the future. Third, the level of saliva pepsin in the morning is likely to reflect the quantity of pepsin refluxed over the last 12 to 24 h. In other words, morning pepsin levels are likely to be associated with the foods and beverages consumed for dinner the day before and their refluxogenic potential [22,26]. Therefore, it is important to advise patients to maintain their normal daily and food habits on the day before the examination. More precisely, they should be instructed not to fast or indulge on the day before the saliva examination.

A limitation of our study is the relatively small number of included patients. Moreover, the low number of patients with LPR-negative results on MII-pH makes the results of some statistical results (particularly, specificity of the Peptest) inconclusive. In addition, the keeping of patients on regular diet regimen before saliva collection for Peptest examination can cause some bias and was discussed above. In addition, patient selection bias cannot be completely reduced when performing studies on LPR. Furthermore, a limitation of all current studies on LPR, including ours, is the lack of a gold standard for the diagnosis of LPR and a consensus on how many LPR reflux episodes are pathological. We attempt to reduce this kind of bias using two thresholds of hypopharyngeal reflux events in analysis. 

Despite its limitations, our study results suggested that a positive Peptest should be considered a strong indication that the patient has LPR, and the patient should be treated accordingly. Therefore, the Peptest could serve as a good screening test for physicians that cannot visualize the larynx, such as general practitioners, allergists, gastroenterologists, and chest physicians, in cases that do not require an upper gastrointestinal endoscopy or bronchoscopy. Another advantage of using the Peptest in the COVID-19 era is its non-invasive nature when compared with flexible laryngoscopy or MII-pH testing. In contrast, a negative Peptest cannot rule out LPR. Therefore, when the Peptest is negative, the patient should be examined by otolaryngologist with the RFS, which has 57% sensitivity according to our findings, or the Reflux Sign Assessment, which is expected to have greater sensitivity [8].

To obtain more robust data and more conclusive results on the Peptest’s diagnostic value, future studies on the subject should be of multicentric nature. Important information regarding whether the Peptest can predict a response to treatment (diet, lifestyle modification, and medication) is needed. This point should be studied in the close future.

## 5. Conclusions

A positive Peptest is very specific for diagnosing LPR, whereas a negative Peptest cannot exclude LPR. Therefore, the Peptest could serve as a screening test for physicians who cannot visualize the larynx, for whatever reason.

## Figures and Tables

**Figure 1 jcm-10-02996-f001:**
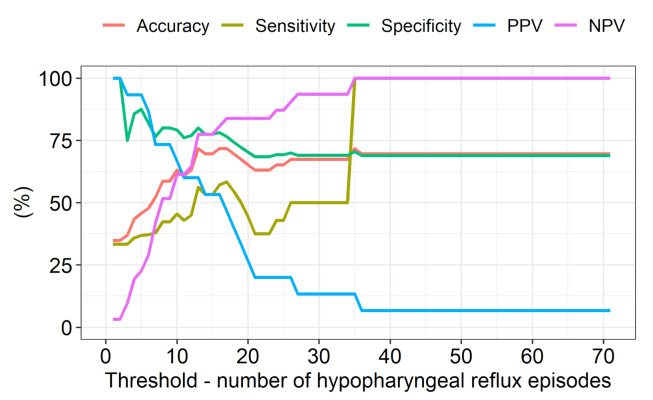
Evolution of the accuracy, sensitivity, specificity, positive predictive value (PPV), and negative predictive value (NPV) of the Peptest when the threshold for a laryngopharyngeal reflux diagnosis was changed incrementally from 1 to 70 episodes.

**Table 1 jcm-10-02996-t001:** Characteristics of the study group.

Characteristic	Peptest (+) (*n* = 15)	Peptest (−) (*n* = 31)	Total (*n* = 46)	*p*
Age (years)	48 (38–64)	49 (37–62)	49 (36–62)	0.953
BMI (kg/m^2^)	23.7 (23.2–26.6)	24.2 (21.8–27.7)	24.2 (22.1–27.1)	0.699
Sex				
female	10 (67)	20 (65)	30 (65)	>0.999
male	5 (33)	11 (35)	16 (35)	

Values are the median (interquartile range) or absolute and relative frequencies (in %), as indicated. *p*-values were evaluated with the Mann–Whitney test or the chi-square test of independence. BMI—Body mass index.

**Table 2 jcm-10-02996-t002:** Diagnostic evaluations of the Peptest, RSI, and RFS in patients with (+) or without (−) a laryngopharyngeal reflux diagnosis based on a threshold of one hypopharyngeal reflux episode.

Parameter	Peptest (+)	Peptest (−)	RSI (+)	RSI (−)	RFS (+)	RFS (−)
LPR (+)	15	30	18	27	26	19
LPR (−)	0	1	1	0	1	0
Accuracy	35(21.4; 50.2)		39 (25.1; 54.6)		57 (41.1; 71.1)	
Sensitivity	33 (20.0; 49.0)		40 (25.7; 55.7)		58 (42.2; 72.3)	
Specificity	100 (2.5; 100.0)		0 (0.0; 97.5)		0 (0.0; 97.5)	
PPV *	100 (78.2; 100.0)		95 (74.0; 99.9)		96 (81.0; 99.9)	
NPV	3 (0.1; 16.7)		0 (0.0; 12.8)		0 (0.0; 17.6)	

Values are the number of patients or diagnostic measure with the 95% confidence interval, as indicated. RSI—Reflux Symptom Index, RFS—Reflux Finding Score, LPR—laryngopharyngeal reflux, PPV—positive predictive value, NPV—negative predictive value. *—important result.

**Table 3 jcm-10-02996-t003:** Diagnostic evaluations of the Peptest, RSI, and RFS in patients with (+) or without (−) a laryngopharyngeal reflux diagnosis based on a threshold of six hypopharyngeal reflux episodes.

Parameter	Peptest (+)	Peptest (−)	RSI (+)	RSI (−)	RFS (+)	RFS (−)
LPR (+)	13	22	13	22	20	15
LPR (−)	2	9	6	5	7	4
Accuracy	48 (32.9; 63.1)		39 (25.1; 54.6)		52 (36.9; 67.1)	
Sensitivity	37 (21.5; 55.1)		37 (21.5; 55.1)		57 (39.4; 73.7)	
Specificity	82 (48.2; 97.7)		46 (16.7; 76.6)		36 (10.9; 69.2)	
PPV *	87 (59.5; 98.3)		68 (43.4; 87.4)		74 (53.7; 88.9)	
NPV	29 (14.2; 48.0)		19 (6.3; 38.1)		21 (6.1; 45.6)	

Values are the number of patients or diagnostic measure with the 95% confidence interval, as indicated. RSI—Reflux Symptom Index, RFS—Reflux Finding Score, LPR—laryngopharyngeal reflux, PPV—positive predictive value, NPV—negative predictive value. *—important result.

**Table 4 jcm-10-02996-t004:** Evaluation of the Peptest for the diagnosis of gastroesophageal reflux disease in patients with (+) or without (−) gastroesophageal reflux disease based on the DeMeester score.

Parameter	Peptest (+)	Peptest (−)
DeMeester score (+)	6	16
DeMeester score (−)	9	15
Accuracy	46 (30.9; 61.0)	
Sensitivity	27 (10.7; 50.2)	
Specificity	63 (40.6; 81.2)	
PPV	40 (16.3; 67.7)	
NPV	48 (30.2; 66.9)	

Values are the number of patients or diagnostic measure with the 95% confidence interval, as indicated. PPV—positive predictive value, NPV—negative predictive value.

## Data Availability

The data presented in this study are available on request from the corresponding author.

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
