# Peer review of "Diagnostic Value of the PeptestTM in Detecting Laryngopharyngeal Reflux"

_jcm, 2021, doi:10.3390/jcm10132996_

Round 1
Reviewer 1 Report
This is an interesting study aiming at evaluating the diagnostic value of the Peptest for detecting laryngopharyngeal reflux (LPR). The conclusion of the Authors is that the peptest had a high diagnostic value as it demonstrated a high specificity and PPV; on the other hand , a negative test could not exclude LPR. One limitation is represented by the relatively low number of patients studied (46) and this should be considered in in the conclusion by the Authors. Nevertheless, I consider this test a valuable additional test for the diagnosis of LPR as a distinct diagnosis from GERD. Recently, Lechien et al. demonstrated that the saliva pepsin concentration of the 24 hour period of testing was significantly with foods and beverages consumed during the testing period and the evening dinner. In this regard, the Authors cited this study in the references but they did not discuss this issue in the Discussion and this may be explained by the fact the the Authors did not consider type of feeding the day before the morning pepsin test. The Authors should discuss on it and on the possible bias generated by type of food and beverages. The Authors should update references and discuss on the results of new studies on this topic. However, the Discussion section is too long and the Authors should reduce the length of the text.
Author Response
Dear Editors and Reviewers,
we are grateful for reviewer’s comments and reccomendations, which are very precious for us. Changes were made according to recommendations and are listed below step by step. Changes in re-submitted manuscript are in “tracking mode” in red colour.
Reviewer 1:
One limitation is represented by the relatively low number of patients studied (46) and this should be considered in in the conclusion by the Authors.
Changes were made according to recommendation – Limitation section was added. Page 8, line 538 – Page 9, line 571
Recently, Lechien et al. demonstrated that the saliva pepsin concentration of the 24 hour period of testing was significantly with foods and beverages consumed during the testing period and the evening dinner. In this regard, the Authors cited this study in the references but they did not discuss this issue in the Discussion and this may be explained by the fact the the Authors did not consider type of feeding the day before the morning pepsin test. The Authors should discuss on it and on the possible bias generated by type of food and beverages.
- To clarify methodology of Peptest examination sentence that “Patients were instructed to have their standard food and drink regimen the day before examination and fast overnight.” was added to Methods section. Page 2, lines 143-145
- Discussion part was upgraded according to reviewer’s recommendations. Page 7, lines 491-497
The Authors should update references and discuss on the results of new studies on this topic.
To our knowledge, all latest studies on relation between Peptest and MII-pH were included. To discuss other studies would lead to even longer discussion and it was recommended to reduce it.
However, the Discussion section is too long and the Authors should reduce the length of the text.
Some parts of Discussion were reduced.
Reviewer 2 Report
Comments to the authors
Overall impression
This is an interesting study investigating the diagnostic value of the Peptest in detecting LPR based on MII-pH monitoring. As the authors used two different criteria for diagnosing LPR (i.e. detection of one or six hypopharyngeal reflux episodes), this paper adds new information to the literature.
Title
I would suggest a slightly modified title (e.g. “Diagnostic Value of the Peptest in detecting laryngopharyngeal reflux”).
Abstract
Background: Describe the Peptest in one sentence and indicate a precise aim of your study (e.g. “The aim of the present study was to investigate the diagnostic value of the Peptest in detecting LPR based on MII-pH monitoring”).
Methods: Slightly reduce this section to the most essential.
Results: Use rounded figures in this section and throughout the paper including all tables (e.g. 35% instead of 34.8%).
Conclusions: Indicate a short and precise conclusion according to the aim and results of your study (see below).
Manuscript
General: The authors use “accuracy” or “diagnostic accuracy” and “diagnostic value” as synonyms, which is not the case. Therefore, I suggest to use the term “diagnostic value” throughout the paper instead of “accuracy” when evaluating the different methods (except when really meaning the accuracy of a test). For example: According to your results, the accuracy of the Peptest in detecting LPR was significantly lower than RFS (34.8% vs 56.5%*) and similar to RSI (34.8% vs 39.1%) and in detecting GERD (34.8% vs 45.7%).
Introduction: Clearly state that there is no accepted gold standard for LPR diagnosis so far. Probably, one method alone is not able to do so, as every diagnostic method shows its false positive and false negative results (and MII-pH monitoring is not an exception!). However, this is a general problem of all LPR studies in the past.
Methods: Why were patients scheduled for lung transplantation included in the study? Did they also have symptoms and signs of LPR? In this case, rather leave this information away as it is not really important for this study. Shortly describe what you mean by nonacid (alkaline? neutral?) and mixed LPR and how you determined the severity of LPR by MII-pH monitoring.
Results: Why were 6 patients excluded from the study? Either explain the reason of exclusion or rather leave these subjects away und mention only the 46 included patients. Additionally, there is a discrepancy between Table 1 and the main text concerning the sensitivity of the Peptest (33.3% vs 33.8%), which should be corrected. Generally, I would suggest to use rounded figures throughout the paper including the tables (e.g. 35% instead of 34.8%). Indicate in the tables, which of your results were significant by an asterisk (*). Then, you will realize that only a few of your results are significant due to the low number of participants. Especially the specificity does not differ between the different test though ranging from 100% to 0%. This should at least be mentioned in the discussion section. Additionally, your results for the Peptest in detecting GERD do not significantly differ from LPR (except for PPV), thus your conclusions concerning GERD are not really maintained by your findings. However, several authors showed that GERD and LPR can also occur simultaneously (probably a third of LPR patients). As MII-pH monitoring is able to distinguish between GERD (positive Demester Score) and LPR (one or six hypopharyngeal reflux episodes), the authors should probably differentiate between GERD alone (GERD+ LPR-) and GERD with LPR (GERD+ LPR+) in Table 4 and see, if there is a significant difference or tendency between these two groups. This would further improve the quality of your paper and maintain the conclusion that a positive Peptest is very specific for LPR but not for GERD (alone).
Discussion: The development of your arguments can be easily understood, and there are no discrepancies. However, I miss a short section about the limitations of your study (e.g. low number of participants, patient selection bias, lack of a gold standard for diagnosing LPR) and an outlook for future studies. Correct the sentence in line 194f as mentioned above (“Not surprisingly, we found that the Peptest had a low accuracy for diagnosing GERD” as this is not maintained by your results).
Conclusions: Indicate a precise conclusion according to the aim and findings of your study (e.g. “A positive Peptest is very specific for diagnosing LPR, whereas a negative Peptest cannot exclude LPR”). Remove the last sentence, as this was not an aim of your study and is not supported by your results (“The MII-pH remains the gold standard for diagnosing LPR”).
References: The references in this paper are appropriate.
Abbreviations: The abbreviations used in the paper are appropriate.
Paper Size
The paper size is adequate.
Tables and Figures
The tables and figures are adequate, Table 4 could be extended as mentioned above (GERD+ LPR- vs GERD+ LPR+).
Author Response
Dear Editors and Reviewers,
we are grateful for reviewer’s comments and reccomendations, which are very precious for us. Changes were made according to recommendations and are listed below step by step. Changes in re-submitted manuscript are in “tracking mode” in red colour.
Reviewer 2
I would suggest a slightly modified title (e.g. “Diagnostic Value of the Peptest in detecting laryngopharyngeal reflux”).
Title was changed according to recommendation.
Abstract Background: Describe the Peptest in one sentence and indicate a precise aim of your study (e.g. “The aim of the present study was to investigate the diagnostic value of the Peptest in detecting LPR based on MII-pH monitoring”). Methods: Slightly reduce this section to the most essential.
Results: Use rounded figures in this section and throughout the paper including all tables (e.g. 35% instead of 34.8%). Conclusions: Indicate a short and precise conclusion according to the aim and results of your study (see below).
Abstract was changed according to recommendation of reviewer – see tracking changes in Abstract section.
Manuscript General: The authors use “accuracy” or “diagnostic accuracy” and “diagnostic value” as synonyms, which is not the case. Therefore, I suggest to use the term “diagnostic value” throughout the paper instead of “accuracy” when evaluating the different methods (except when really meaning the accuracy of a test). For example: According to your results, the accuracy of the Peptest in detecting LPR was significantly lower than RFS (34.8% vs 56.5%*) and similar to RSI (34.8% vs 39.1%) and in detecting GERD (34.8% vs 45.7%).
Thank you for this important remark, changes in manuscript were made accordingly.
Introduction: Clearly state that there is no accepted gold standard for LPR diagnosis so far. Probably, one method alone is not able to do so, as every diagnostic method shows its false positive and false negative results (and MII-pH monitoring is not an exception!). However, this is a general problem of all LPR studies in the past.
Introduction section was changed according to reviewer’s recommendations. Page 1, lines 35-37
Methods: Why were patients scheduled for lung transplantation included in the study? Did they also have symptoms and signs of LPR? In this case, rather leave this information away as it is not really important for this study.
Yes, thank you, the information was omitted since it was not important for the purpose of the study.
Shortly describe what you mean by nonacid (alkaline? neutral?) and mixed LPR and how you determined the severity of LPR by MII-pH monitoring.
Methods section was changed according to reviewer’s recommendations. Page 3, line 174 - 181
Results: Why were 6 patients excluded from the study? Either explain the reason of exclusion or rather leave these subjects away und mention only the 46 included patients.
Reason for exclusion was added. Page 4, line 202-203
Additionally, there is a discrepancy between Table 1 and the main text concerning the sensitivity of the Peptest (33.3% vs 33.8%), which should be corrected. Generally, I would suggest to use rounded figures throughout the paper including the tables (e.g. 35% instead of 34.8%).
Typing error was corrected and numbers were rounded throughout the paper.
Indicate in the tables, which of your results were significant by an asterisk (*). Then, you will realize that only a few of your results are significant due to the low number of participants.
Important result (PPV – positive predictive value) was indicated by asterisk in Table 2 and 3
Especially the specificity does not differ between the different test though ranging from 100% to 0%. This should at least be mentioned in the discussion section.
This information was added to limitations of the study. Page 7, lines 538-540
Additionally, your results for the Peptest in detecting GERD do not significantly differ from LPR (except for PPV), thus your conclusions concerning GERD are not really maintained by your findings. However, several authors showed that GERD and LPR can also occur simultaneously (probably a third of LPR patients).
Thank you for this point, information regarding GERD was ommited.
As MII-pH monitoring is able to distinguish between GERD (positive Demester Score) and LPR (one or six hypopharyngeal reflux episodes), the authors should probably differentiate between GERD alone (GERD+ LPR-) and GERD with LPR (GERD+ LPR+) in Table 4 and see, if there is a significant difference or tendency between these two groups. This would further improve the quality of your paper and maintain the conclusion that a positive Peptest is very specific for LPR but not for GERD (alone).
Thank you for this idea, first we were considereing it, but omitted because most of our patients were LPR positive on MII-pH and results would be inconclusive.
Discussion: The development of your arguments can be easily understood, and there are no discrepancies. However, I miss a short section about the limitations of your study (e.g. low number of participants, patient selection bias, lack of a gold standard for diagnosing LPR) and an outlook for future studies.
Section about limitations (Page 7, line 538- Page 8, line 571) and future outlook (Page 8, lines 583-586) was added.
Correct the sentence in line 194f as mentioned above (“Not surprisingly, we found that the Peptest had a low accuracy for diagnosing GERD” as this is not maintained by your results).
Sentence was omitted.
Conclusions: Indicate a precise conclusion according to the aim and findings of your study (e.g. “A positive Peptest is very specific for diagnosing LPR, whereas a negative Peptest cannot exclude LPR”).
Remove the last sentence, as this was not an aim of your study and is not supported by your results (“The MII-pH remains the gold standard for diagnosing LPR”).
Conclusion section was changed and simplified.